# The institutional origins of vaccines distrust: Evidence from former-Soviet countries

**Joan Costa-Font[1]°, Jorge Garcia-Hombrados[2]°, Anna Nicińska[3]°***

**1** Department of Health Policy, London School of Economics and Political Science (LSE), London, United Kingdom, **2** Facultad de Ciencias Economicas, Universidad Autonoma de Madrid, Madrid, Spain, **3** Faculty of Economic Sciences, University of Warsaw, Warsaw, Poland

° These authors contributed equally to this work.
* anicinska@wne.uw.edu.pl

## Abstract

How is vaccines scepticism related to the exposure to Soviet communism? Using individual level evidence on vaccine trust with regards to its efficiency and safety in 122 countries that differ in their exposure to communism, we document that past exposure to Soviet communism is associated with lower trust in vaccination. We show that exposure to socio-political regimes can negatively affect trust in vaccines, which is explained by weak trust in both government and medical advice from doctors as well as in people from the neighbourhood. These results suggest that roots of vaccine scepticism lie in a wider distrust in public and state institutions resulting from the exposure to Soviet communism.

## Introduction

Institutional legacies can influence various preferences, among which the health care preferences are essential for the performance of public health care. The channels through which institutions can affect individual health-related beliefs and behaviours include trust in the effectiveness of health care treatments. One such treatment are protective interventions aiming at stimulating an individual's immune system, such as vaccines. In this paper we examine one particular form of institutional legacy, namely the exposure to Soviet communism, and its repercussions for vaccine skepticism. Exposure to Soviet communism is related to the reaction to historical events (mass vaccinations during Soviet times), weak trust in government and health system, suspicion of large business organizations (such as big pharma) as well as the extent of adoption of egalitarian values [1]. These in turn seem to be relevant for vaccine skepticism.

Previous research documents that exposure to Soviet communism is found to be relevant for the formation of preferences [2, 3] and detrimental to various forms of trust (most importantly political, generalized, and in public institutions) [4–9]. Global differences in vaccination trust (see Fig 1) suggest a similar picture. Poland, Hungary and Russia in 2021 were the three countries with the highest COVID-19 vaccine hesitancy in Europe according to Ipsos survey [10]. The 2018 Wellcome Global Monitor Report (that uses the dataset employed in this study), establishes an association between the trust in doctors and nurses and the trust in vaccines, yet this was weaker in European countries. Some research shows that Eastern Europe

**Data Availability Statement:** Wellcome Trust, The Gallup Organization Ltd. (2019). Wellcome Global Monitor, 2018. [data collection]. 2nd Edition. UK Data Service. SN: 8466, http://doi.org/10.5255/UKDA-SN-8466-2.

**Funding:** A.N., PPN/BEK/2019/1/00039, the the Polish National Agency for Academic Exchange, Bekker programme, https://nawa.gov.pl/. The funders had no role in study design, data collection and analysis, decision to publish, or preparation of the manuscript.

**Competing interests:** The authors have declared that no competing interests exist.

reveals the lowest scores for vaccine confidence compared to other sub-regions [11]. Is such vaccine hesitancy associated to the exposure to Soviet communism?

This paper inquires about what are the effects of the exposure to Soviet communism on the trust in vaccines efficiency and in vaccines safety. We hypothesize that exposure to communism is associated with increased vaccine skepticism, specifically individuals with larger exposures to communism have lower trust both in vaccines' safety and in their efficiency. Using the Wellcome Global Monitor (WGM) dataset, we examine whether exposure to communism explains different dimensions of vaccine trust. This is important in the light of evidence showing that a country's individualism is associated with more COVID-19 deaths [12], which might have nurtured the idea that vaccines are primarily benefiting countries with more individualistic societies. This is especially relevant as the root causes of COVID-19 vaccination hesitancy are common to vaccine hesitancy more generally [13], and they include lack of trust in the health system and barriers to access to health care, as well as misinformation. In addition to luck of trust, in this paper we argue that some historical legacies related to exposure to Soviet communism (such as experiences of mass vaccinations) can also be the source of vaccine scepticism.

Hesitant individuals often ignore that vaccines are protective interventions that can have a long-lasting impact on our health, and vaccination is a pro-social behaviour as it helps to protect others. However, if some shares of the population refuse to vaccine, it compromises herd immunity objectives, which is the core of the COVID-19 recovery strategy in almost every Western country. Consistently, risk seeking and less pro-social individuals [14] are less likely

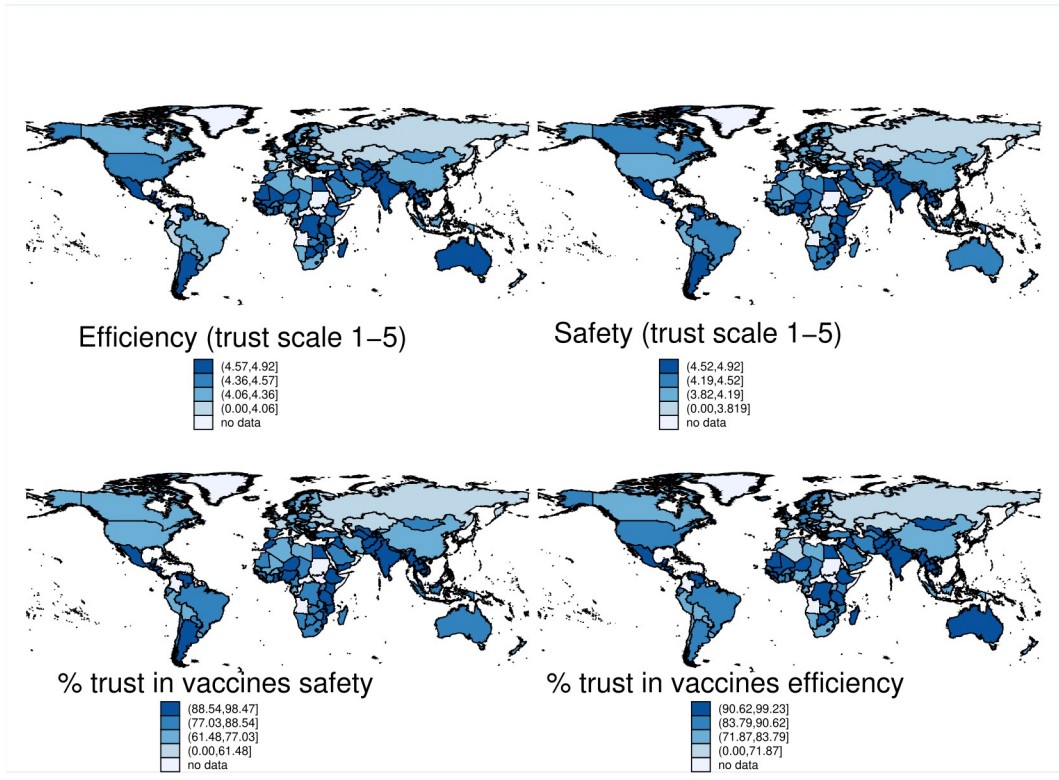

**Fig 1. Trust in vaccines efficiency and safety in the countries of the world.** *Notes*: The above maps were prepared using resources on country boundaries released by the World Bank and available in the public domain. The sample is split in quantiles according to the value of the variable. The colour indicates the quantile as shown in the legend. *Source*: WGM 2018.

to vaccinate, and women are generally less likely to support vaccinations than men [15]. Similarly, limited knowledge and wider family opposition play an important role in cross-country vaccine hesitancy [16].

Compliance with vaccination plans is a barrier to overcome. For instance, the anti-H1N1 vaccine during the 2009 influenza pandemic was low [17]. The World Health Organization (WHO) already in 2019 identified vaccine hesitancy as one of the top ten global health threats, which they define as the delay in acceptance or refusal of vaccination despite availability of vaccination services. Some of the technology drivers of vaccination resistance such as the role of social media are well known [18]. Nonetheless, the origin of vaccination resistance can be traced to other mechanisms that explain the divide between Eastern and Western Europe [12, 19].

The role of trust in the health system, and the health care system more generally, is fundamental. Vaccines are mostly developed in Western Europe, which might not always be appreciated in some Eastern European countries. While previous studies have documented the role of factors such as religiosity or spirituality on attitudes towards science [20, 21], the development of populism is connected to vaccine hesitancy [22, 23]. Indeed, many Eastern European countries have exhibited weaker health system reaction to COVID-19 which is found to drive vaccine attitudes. For instance, evidence from the United Kingdom documents that people residing in locations where intensive care units were under stress in the first wave of the COVID-19 pandemic are found to be more vaccine hesitant [24]. However, the role of trust is found to be negatively correlated with vaccination intentions [25]. Consistently with the effect of forced mass vaccinations during Soviet times, Schmelz and Bowles [25] find that vaccination enforcement crowds out voluntary commitment. Hence, opposition to mass vaccination during Soviet times might explain vaccination reluctance. Mass vaccinations were a pillar of public health care systems widely used in former Soviet Union (Union of Soviet Socialist Republics) and allied countries [26]. Hungary managed to set the benchmark model used by the WHO to fight for polio, and Czechoslovakia was among the first countries to eradicate the disease [27]. Communist officials contrasted the polio-free world of Eastern Europe with struggling Western nations [28]. However, the military-like organisation of vaccinations and its compulsory participation [29] led to a questioning of such campaigns after transition [28].

That said, exposure to Soviet communism might nurture conspiracy theories that increase vaccine hesitancy [1]. Stronger "conspiracy thinking" driven by the distrust towards official accounts during Soviet times might explain vaccination reluctance [30]. The rest of the paper will discuss the data and methods used and the results of our analysis.

## Data and methods

### Data

We refer to the recent pre-pandemic data on trust in general in vaccines' safety and efficiency. The data used in the analysis comes from the WGM 2018 survey, which yields representative sample of adults from 122 countries with information on attitudes towards science and health challenges, and trust in science and health professionals in particular. In addition, it provides baseline socio-economic characteristics of respondents. We refer to the data on trust in vaccines efficiency and safety, asked in the two following questions: "Do you agree, disagree, or neither agree nor disagree with the following statement?": "Vaccines are efficient."; "Vaccines are safe". Respondents who agreed or disagreed with the statements, were asked how strong are their opinions, which yielded a 5-point scale "strongly agree", "somewhat agree", "neither agree nor disagree", "somewhat disagree", "strongly disagree". The answers are re-coded in our study so that the higher score denotes the higher trust in vaccines.

Measures of general confidence in medical authorities and their advice, as well as generalized trust and confidence in government observed in our data, provide interesting context to the trust in vaccination. We use questions pertaining to trust in "hospitals and health clinics" measured on binary scale and in "doctors and nurses in this country" measured on 4-point scale. Two separate questions specifically ask about trust in "medical and health advice from the government in this country" and "from medical workers, such as doctors and nurses, in this country? A lot, some, not much, or not at all?". Generalized trust in neighbours and in government are measured on 4-point scale using respectively the two following questions: "How about the people in your neighborhood? Do you trust them a lot, some, not much, or not at all?"; "How about the national government in this country? Do you trust them a lot, some, not much, or not at all?"

S1 Table provides descriptive statistics of these measures. A substantial portion (15%) of the examined population was exposed to Soviet communism despite the study took place 27 years after the fall of the Berlin Wall and the average age of respondents was about 43 years. We measure the number of years individuals lived under communism ranging from 0 to 70, to find that globally average respondent lived for 3.7 years under Soviet communism.

Fig 2 (for more details see S2 Table) reports the dates of entry to and exit from the Soviet communism regime. We use the date of the introduction of the socialist constitution to the country and first free elections as the mark of the entry to and exit from the Soviet communism, respectively. In addition to the length of the exposure to Soviet communism, we construct a dummy variable measuring if the exposure took place or not. Because the WGM data do not allow for distinction between communist and non-communist parts of Germany, we decided to exclude Germany from the analysis. Similarly, we restrict our analysis by excluding countries exposed to other than Soviet (Marxist-Leninist) types of communism (Burkina Faso, Chad, China, Congo, Egypt, Ghana, Guinea, Iraq, Laos, Libya, Madagascar, Mali, Mauritania, Nicaragua, Vietnam, Venezuela, Senegal, Sierra Leone, Tunisia, Zambia). Despite Soviet communism shares a number of similarities with other authoritarian regimes, it remains distinct from other communist regimes [31] and we cannot test in the present study whether other forms of authoritarian regimes or forms of communism could have the same effect.

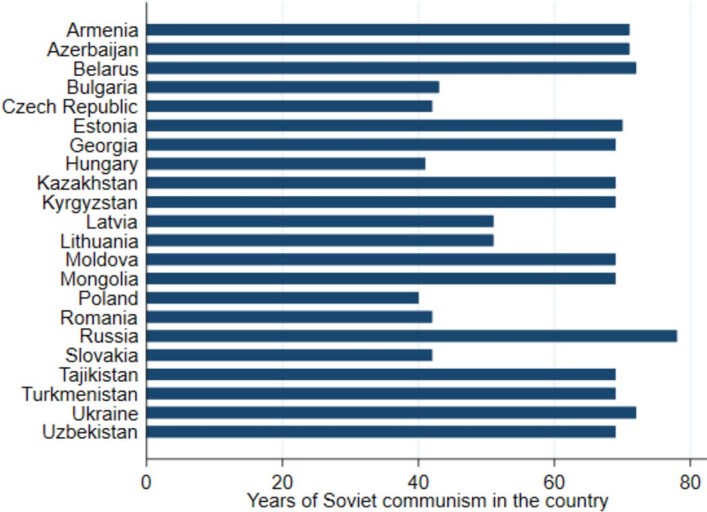

**Fig 2. Number of years countries were ruled by Soviet communist regimes.** *Notes*: The above figure shows the number of years each country was ruled by a Soviet communist regime and *not* the average number of years respondents in a country lived under the Soviet rule. For the latter, see S2 Table.

## Identification strategy

To estimate the effect of exposure to communism on our measures of trust, we estimate the following regression:

$$
\begin{aligned}
Trust\ in\ Vaccines_{ict} = \quad & \beta_0 + \beta_1 Exposure\ to\ Communism_{ict} + Year\ of\ birth\ FE_{ic} \\
& + Country\ FE_c + Country - specific\ trends_{ct} + X_{ict} \qquad (1) \\
& + \varepsilon_{ict}
\end{aligned}
$$

where *Trust in Vaccines$_{ict}$* measures the degrees of trust in the safety and effectiveness of vaccines of individual $i$, living in country $c$ and born in year $t$. *Exposure to Communism$_{ict}$* measures the degree of exposure to communism of individual $i$ which is determined by country of residence $c$ and year of birth $t$. We measure exposure in two different ways: either as the number of years lived under a communist regime or as a dichotomous variable equal to 1 if individual lived at some point during life in a communist regime. Parameter $\beta_1$ yields the effect of the exposure to communism operationalized either as a dummy for being exposed to communism (0–1) or as the continuous number of years one was exposed to communism (0–70 years). The specification includes year of birth and country of residence fixed effects, country-specific time trends and a set of control variables $X_{ic}$ including gender and, depending on the particular specification, dummy for urban or rural area, having children, religiosity, and education levels (primary, secondary, tertiary). $\varepsilon_{ict}$ is the error term and standard errors are clustered at the country level.

## Results

### Trust in vaccines

The data shown in Fig 1 depict the vaccine sceptisicm, specifically the trust in vaccines' efficiency and vaccines' safety at a country level. Geographical regions formerly belonging to or aligned with the Soviet Union stand out from the rest of the world. The trust in vaccination in these regions seems to be the lowest in the world, despite the fact that individuals in vast majority agree with the statements that vaccines are safe and efficient.

We document (cf. Fig 3) that the exposure to Soviet communism is associated with the increased vaccine scepticism. We find negative effects on trust in vaccines' efficiency as well as on their safety. Controlling for year of birth, gender, country of residence and time trends specific to the country of residence, we find that individuals ever exposed to Soviet communism report lower by 2 per cent on average trust in vaccines efficiency measured on 5-point scale. The decrease due to exposure to communism is the same (2 per cent) for the trust in vaccines safety. The effects size is positively correlated with the length of the exposure to communism.

### Underlying factors

Different factors might explain why exposure to Soviet communism is associated with low trust in vaccines. One potential factor is the negative effect of communism on trust in institutions documented in different studies [4–9], which might mediate the negative association between exposure to communism and low vaccination rates in some communist countries [32].

We examine the feasibility of the trust hypothesis through exploring the association between Soviet communism and trust in different institutions. The results are reported in Fig 4. We find that individuals who lived under Soviet communism are significantly less likely to trust in health advice given by their governments and by medical doctors in their country. Furthermore, the results confirm that exposure to communism is associated with reduced

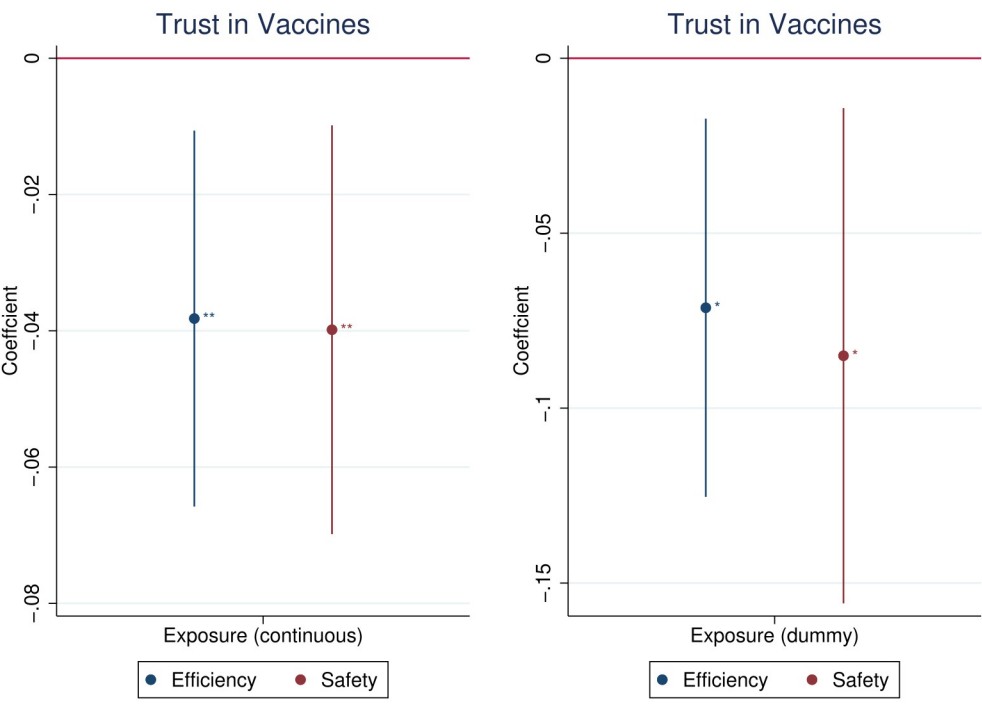

**Fig 3. Effects of the exposure to Soviet communism on trust in vaccines' efficiency and safety.** *Notes*: The treatment variable is the length of exposure to Soviet communism in years smoothed with inverse hyperbolic sine function. Control variables: age fixed-effects, gender, country fixed-effects and country-specific time trend. Presented confidence intervals at 95% significance level are obtained using standard errors clustered by country. Statistical significance: $* - p < 0.10$, $** - p < 0.05$, $*** - p < 0.01$. *Source*: WGM 2018.

generalized interpersonal trust and confidence in government. The reduction of government trust is twice as big as the reduction of interpersonal trust for individuals with any exposure to Soviet communism, indicating particular importance of state institutions. While they are not conclusive, these results are consistent with the hypothesis that the association between exposure to communism and trust in vaccines could be partially explained by the effect of communism on trust in health and other institutions documented in previous studies.

The preferences can be transmitted over generations [33]. However, the inter-generational transmission would lead to the underestimation of the exposure to communism effects. The fact that we find an effect of the number of years of exposure to communism and not only of whether the individual is exposed or not suggests that documented effects are not driven by the transition to free-markets nor inter-generational transmission, but indeed by the exposure to Soviet communism.

## Robustness

Robustness of our main results to various types of Soviet communism implementations is reassuring. We examined the effects of exposure to communism on trust in vaccines (efficiency and safety) in the selected country groups, specifically those belonging to Soviet Union, those affected by Soviet communism except for Russia, and those affected by Soviet communism except for Baltic and Western Republics of the Soviet Union, as shown in detail in Fig 5. In addition, we confirm that the results are robust to controlling for the predominantly Orthodox Christian religion in a country (cf. S5 Table). Furthermore, we examined a number of alternative set of control variables (including living in rural or urban area, having children, individual

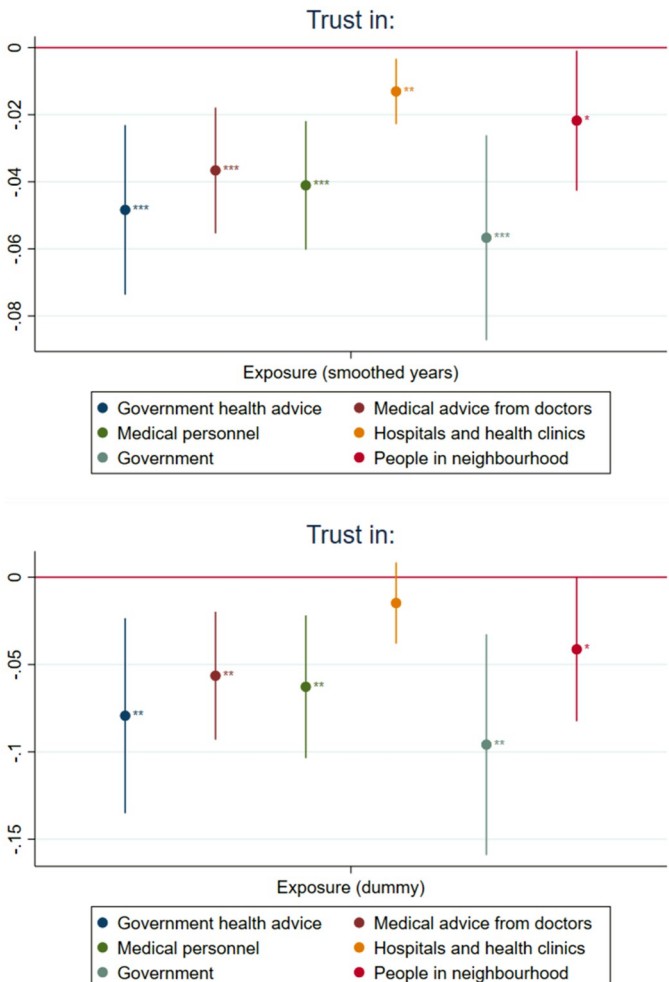

**Fig 4. Effects of the exposure to Soviet communism on generalized trust and trust in medical care.** *Notes*: The treatment variable is the length of exposure to Soviet communism in years smoothed with inverse hyperbolic sine function. Control variables: age fixed-effects, gender, country fixed-effects and country-specific time trend. Presented confidence intervals at 95% significance level are obtained using standard errors clustered by country. Statistical significance: $* - p < 0.10$, $** - p < 0.05$, $*** - p < 0.01$. *Source*: WGM 2018.

religiosity, and educational attainment) to find the results robust to model specifications, details of which are presented in the S1 Fig.

## Discussion and conclusion

Ensuring a swift vaccination is an essential part of disease prevention, which has become more important in the context of a pandemic, given its wide economic and social consequences. Consistently, vaccination has been a core part of the COVID-19 recovery strategy in almost every country. Although COVID-19 vaccination trials indicate the available vaccines are safe and produce the expected effects on the immune response, a significant share of the population is unwilling to take the vaccine.

### Vaccine scepticism

As in most vaccine-preventable diseases, vaccination resistance is common among groups of individuals who a) refuse to vaccinate on religious and political grounds, and in some cases

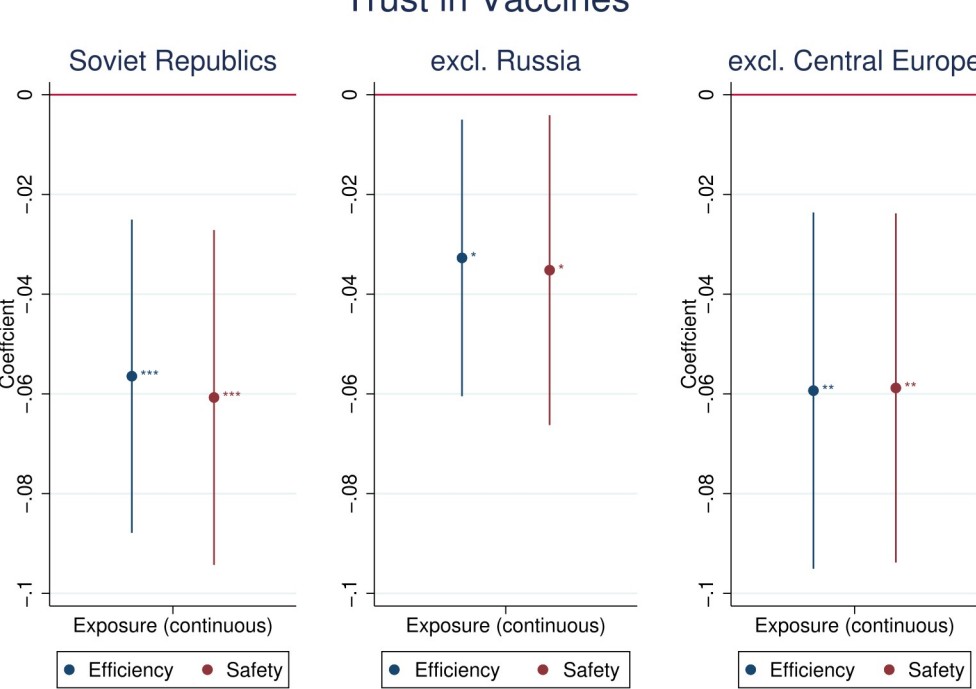

**Fig 5. Robustness of the main results to country groups exposed to Soviet communism.** *Notes*: Soviet Republics: Armenia, Azerbaijan, Belarus, Estonia, Georgia, Kazakhstan, Kirghistan, Latvia, Lithuania, Moldova, Russia, Tajikistan, Turkmenistan, Ukraine, Uzbekistan. Central Europe: Belarus, Czech. Republic, Estonia, Georgia, Hungary, Latvia, Lithuania, Poland, Romania, Slovakia, Ukraine. The treatment variable is the length of exposure to Soviet communism in years smoothed with inverse hyperbolic sine function. Control variables: age fixed-effects, gender, country fixed-effects and country-specific time trend. Presented confidence intervals at 95% significance level are obtained using standard errors clustered by country. Statistical significance: $^{*}$—$p < 0.10$, $^{**}$ − $p < 0.05$, $^{***}$ − $p < 0.01$. *Source*: WGM 2018.

even with some scientific backing [34], and b) are 'vaccine hesitant' [35], either due to their limited knowledge of vaccine side effects or a more general distrust of the health system, which is sensitive to experiences of mass vaccination, and past health care misinformation. Slow vaccination behaviour is often driven by the role of misinformation of their risks, and/or by the formation of conspiracy thinking around the limited benefits [19], or even concerns about the value of vaccination [36] and its side effects [15]. Consistently, evidence from an Australian attitudes survey on COVID-19 finds that about 86% respondents reported they intended to get the vaccine, and almost half (44%) of those who would not were more likely to believe the threat of COVID-19 has been exaggerated [37]. Sherman et al. [38] show that in July 2020 about 64% of the UK population were willing to be vaccinated when the COVID-19 vaccine became available.

## Regaining trust

A central determinant of vaccine hesitancy refers to limited trust in medicine and the health system, which varies significantly among certain groups and between countries and cultures, and more generally to limited public trust [25, 39] present in the "low-trust societies" [4] and post-communist countries [40]. Indeed, given that knowledge about vaccination is generally limited, individual decisions tend to rely on trust [41]. An environment of distrust in institutions and (medical) "experts" can hamper the public acceptability of vaccines [42]. Some

socio-political regimes (Soviet communism) are found to exhibit low trust [6]. Consistently, this paper shows that individuals' vaccines scepticism is related to the exposure to a specific socio-political regime, namely the exposure to Soviet communism, that is shown to have reduced trust in public institutions [6–9], and forced mass vaccinations using military like organisations [43] would have influenced on people's positional attitudes to taking up the vaccines. Finally, such differences reflect in a clear East and West European divide in vaccine update [44].

## Lessons

One lesson that emerges from this evidence is that compulsory vaccinations might backfire as they might remind individuals of their Soviet legacies and they have been found to crowd out voluntary commitment to vaccinate [25]. In contrast, actions undertaken by international independent bodies as well as the way in which governmental vaccination programs are implemented by local authorities or even, non-profit organisations, can reduce the effects of government distrust enhancing vaccine scepticism. Hence, in promoting vaccination in Eastern Europe, one strategy is to attract other types of stakeholders that are perceived as more trustworthy than the state and local elites.

## Supporting information

**S1 Table. The descriptive statistics of the WGM 2018 research sample employed in the analysis.**
(PDF)

**S2 Table. The specific dates of entry to and exit from the Soviet communism for all formerly communist countries examined in the analysis.**
(PDF)

**S3 Table. The details of the estimates presented in Fig 3.**
(PDF)

**S4 Table. The details of the estimates presented in Fig 4.**
(PDF)

**S5 Table. Robustness check controlling for Orthodox Christian countries.**
(PDF)

**S1 Fig. Robustness check employing four alternative sets of controls in the multi-variate analysis.**
(PDF)

**S1 Checklist. STROBE checklist.** This PDF file contains STROBE check-list for analysis using cross-section data.
(PDF)

## Acknowledgments

All errors are our own and the usual disclaimer applies. We are grateful to Anat Rosenthal, Joanna Rachubik and two anonymous referees for their valuable comments.

## Author Contributions

**Conceptualization:** Joan Costa-Font, Jorge Garcia-Hombrados, Anna Nicińska.

**Investigation:** Joan Costa-Font, Jorge Garcia-Hombrados, Anna Nicińska.

**Methodology:** Joan Costa-Font, Jorge Garcia-Hombrados, Anna Nicińska.

**Project administration:** Joan Costa-Font, Anna Nicińska.

**Resources:** Jorge Garcia-Hombrados, Anna Nicińska.

**Software:** Anna Nicińska.

**Supervision:** Joan Costa-Font, Anna Nicińska.

**Validation:** Joan Costa-Font.

**Visualization:** Jorge Garcia-Hombrados, Anna Nicińska.

**Writing – original draft:** Joan Costa-Font, Jorge Garcia-Hombrados, Anna Nicińska.

**Writing – review & editing:** Joan Costa-Font, Jorge Garcia-Hombrados, Anna Nicińska.

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
