## [Decision Letter · Decision Letter 0]

2 Jul 2021

PONE-D-21-12657

The institutional origins of vaccines distrust: Evidence from former-communist countries

PLOS ONE

Dear Dr. Nicińska,

Thank you for submitting your manuscript to PLOS ONE. After careful consideration, we feel that it has merit but does not fully meet PLOS ONE’s publication criteria as it currently stands. Therefore, we invite you to submit a revised version of the manuscript that addresses the points raised during the review process.

The submission has now been reviewed and I have received the referees’ evaluation of your paper. As you can see, the reviewers find the theme of your manuscript interesting but point to shortcomings and weaknesses that need to be addressed and remedied. Should you be willing to undertake them, I will accept for further consideration a substantially revised version of your manuscript that carefully and meticulously takes onboard  the reviewers’ recommendations and suggestions addressing all the issues raised in the assessment reports appended below. When resubmitting please indicate how the revised version addresses the referees’ concerns and concomitant suggestions.

We look forward to receiving your revised manuscript.

Kind regards,

Athina Economou

Academic Editor

PLOS ONE

Journal Requirements:

4. As part of your revision, please complete and submit a copy of the STROBE checklist, a document that aims to improve experimental reporting and reproducibility of observational studies for purposes of post-publication data analysis and reproducibility: https://www.strobe-statement.org/?id=available-checklistsPlease include your completed checklist as a Supporting Information file. Note that if your paper is accepted for publication, this checklist will be published as part of your article.

6. We note that Figure 1 in your submission contain map images which may be copyrighted. All PLOS content is published under the Creative Commons Attribution License (CC BY 4.0), which means that the manuscript, images, and Supporting Information files will be freely available online, and any third party is permitted to access, download, copy, distribute, and use these materials in any way, even commercially, with proper attribution. For these reasons, we cannot publish previously copyrighted maps or satellite images created using proprietary data, such as Google software (Google Maps, Street View, and Earth). For more information, see our copyright guidelines: http://journals.plos.org/plosone/s/licenses-and-copyright.

6.1.    You may seek permission from the original copyright holder of Figure 1 to publish the content specifically under the CC BY 4.0 license. 

62.    If you are unable to obtain permission from the original copyright holder to publish these figures under the CC BY 4.0 license or if the copyright holder’s requirements are incompatible with the CC BY 4.0 license, please either i) remove the figure or ii) supply a replacement figure that complies with the CC BY 4.0 license. Please check copyright information on all replacement figures and update the figure caption with source information. If applicable, please specify in the figure caption text when a figure is similar but not identical to the original image and is therefore for illustrative purposes only.

Reviewers' comments:

Reviewer's Responses to Questions

**Comments to the Author**

1. Is the manuscript technically sound, and do the data support the conclusions?

Reviewer #1: Partly

Reviewer #2: Partly

Reviewer #3: Yes

2. Has the statistical analysis been performed appropriately and rigorously? 

Reviewer #1: Yes

Reviewer #2: I Don't Know

Reviewer #3: Yes

3. Have the authors made all data underlying the findings in their manuscript fully available?

Reviewer #1: Yes

Reviewer #2: Yes

Reviewer #3: Yes

4. Is the manuscript presented in an intelligible fashion and written in standard English?

Reviewer #1: Yes

Reviewer #2: Yes

Reviewer #3: Yes

5. Review Comments to the Author

Reviewer #1: In the current manuscript, entitled “The institutional origins of vaccines distrust: Evidence from former-communist countries”, the authors show the association between being exposed to Soviet communism affect trust in vaccines, which is explained by weaker institutional and governmental trust. I enjoyed reading the manuscript as it was clearly written, and deals with what I think is an extremely important topic, namely, vaccine hesitancy. However, I believe that there are two major issues that should be addressed. I list these below, and I hope that the author would find them useful.

1. I think that some more elaboration on why exposure to Soviet communism would be relevant to vaccine hesitancy is warranted. Is it something about the history of those countries? Is it something about the ideology? Culture? For example, recent research by Maaravi and colleagues shows the relationship between individualism / collectivism and COVID-19 spread (which I can only assume is correlated with exposure to Soviet communism, although they show a trend that to some extent is opposite to the current research results). Other unpublished research by Adam-Troian and colleagues (see here https://psyarxiv.com/nzg7x) examines the association between collectivism, masculinity (controlling for other relevant variables such as Human Development Index) to belief in conspiracy theories (which was linked to institutional trust and vaccine hesitancy). All of this to say that I think that some more precision about what is it exactly in exposure to Soviet communism that leads to vaccine hesitancy and lower levels of institutional and governmental trust is important.

2. I believe that in order for the analysis to be persuasive it has to control for other relevant variables that are related to institutional, governmental and vaccine trust (both on the country and individual levels). Both the work by Adam-Troian et al. and Maaravi et al. I mentioned above control for some relevant such measures.

3. The results on “psychological mechanisms” go some way to address my first comment, but I think that there’s still a need to further explain what is it in the exposure to Soviet communism that drives these effects. Is it something about the type of regime? Is it something about the process in which these regimes collapsed and what happened since that broke individuals’ trust? Is the trend only relevant to Soviet communism? Why would that be the case? Is it because it collapsed? Is it just because vaccination was obligatory in the former Soviet Union? In other words, I think that more is needed to be done in order to explain whether the trends the authors report in the paper are specific to Soviet communism, and if so, why? Or whether it is something that has to do with regime changes or authoritarian regimes, etc.

4. As a minor comment, I think that it would be good to label the y-axis in the figures. I am not sure it is clear.

5. As another minor comment, maybe some more explanation about how exposure to communism was computed in the model. The authors indicate that it was measured in two different ways, but then it was less clear to me how these two different ways were combined.

Reviewer #2: When I saw the Asbtract for this paper, I was very excited. I undertake sociological research on 'trust in vaccinations', so to see a potential paper exploring the impact of communism on trust in vaccinations was exciting. The authors use exisitng data from the Wellcome Global Monitor (which is fine) and undertake some statistical analysis (which I am not equipped to review). The Introduction is too broad, and does not adequately present and critique global literature on 'trust in vaccinations' and needs to enagge much more deeply with the sociology of trust. There is a huge litertaure on trust in governments, medical institutions, pharamceutical industries etc - very little if any of this has been engaged with. The authors need to be able to set their study and findings within these literatures - so they can identify 'what we already know' and 'what's new'. On lines 24-26, there's a referencing problem which goes throughout the paper (the first references are 6 and 1? In the Discussion, the authors could/should engage with sociologists who have explored how trust differs between different societies - for example, Fukuyama explored trust in different societal structures. In the Discussion, the parapgraph on lines 151-161 does not seem to be relevant at all - the authors have introduced COVID-19 context, but the Wellcome data predated COVID-19.

Reviewer #3: Title: The institutional origin of vaccine distrust: Evidence from former-communist countries

Journal: Plos One

The article is a study of vaccine distrust in former-communist countries. Based on an analysis of data from the Welcome Global Monitor (WGM) database, the authors seek to show that exposure to communist regimes play an important role in distrust in public institutions that leads to vaccine hesitancy. Considering the scale of the challenge associated with vaccine hesitancy, especially during a pandemic, this article’s contributions to policy framing and guidelines are timely and necessary.

While the article does offer an important and timely exploration of vaccine hesitancy in former-communist countries, there is still some work to be done in establishing a coherent narrative, and tightening the arguments by addressing not only the results of the analysis but also more of the literature on trust, hesitancy and health policy. Addressing these issues (and a few other minor comments) will strengthen the arguments made in the article. More specific comments follow:

1. The authors provide very clear and admirable inclusion/exclusion criteria for which they should be congratulated. However, the grouping of former-communist counties and the assumption that all communist experiences are similar might be problematic. While it might be difficult to account for the broad variety of experiences based on the available database, I suggest the authors address this methodological difficulty.

2. On page 3, the authors claim that the “military-like organizations and its compulsory participation lead to a questioning of such campaigns after transition”. While these claims might be intuitive to scholars of former-communist health systems, many of the journal’s readers may not be acquainted with this argument. Therefore, it would be helpful to add sources baking the claim.

3. In the Data section (page 4) the authors address the length of the influence of life under communist regimes. If possible, it would be helpful to address inter-generational effects as well.

4. In their analysis of Underlying mechanisms (page 6) the authors claim that trust has deteriorated. This framing assumes that trust existed in the past, however it’s unclear in the text if this is indeed the case.

5. The article has the potential to contribute greatly to the discussion on trust in state institutions and vaccine hesitancy. However, in order to do so more attention is needed to the already broad body of knowledge on the issue. The theorizing of trust and hesitancy appears only in the discussion, and it would be helpful to address it earlier in the article.

6. Along the same line, a clearly framed discussion on trust in institutions and hesitancy would help the authors take their place in a very active field (and explain how is this case different from other cases).

7. I suggest the authors look at Heidi Larson’s book Stuck (2020) on vaccine rumors. While other works by Larson are cited, this might be an interesting read in terms of framing the context.

8. On page 8, the authors shift to COVID-19. While this is obviously a burning issue (and of relevance to the article) it is not necessarily clear how a study conducted pre-COVID-19 relates to the matter at hand. A clearer connection should be made in order for this important section to be a better fit.

9. The authors end with a very interesting note on the use of compulsory mechanisms. I encourage the authors to be even bolder in their conclusions, and odd their recommendations.

10. As someone who is interested in trust in the state and health policy I have truly enjoyed reading this article. I thank the authors for their analysis.

For the reasons mentioned above I recommend accepting this article for publication pending revisions. I look forward to seeing this important work in print.

Kind regards

6. PLOS authors have the option to publish the peer review history of their article (what does this mean?). If published, this will include your full peer review and any attached files.

Reviewer #1: No

Reviewer #2: No

Reviewer #3: **Yes: **Anat Rosenthal

---

## [Author Response · Author response to Decision Letter 0]

20 Sep 2021

Report to the referees

Reviewer #1: In the current manuscript, entitled “The institutional origins of vaccines distrust: Evidence from former-communist countries”, the authors show the association between being exposed to Soviet communism affect trust in vaccines, which is explained by weaker institutional and governmental trust. I enjoyed reading the manuscript as it was clearly written, and deals with what I think is an extremely important topic, namely, vaccine hesitancy. However, I believe that there are two major issues that should be addressed. I list these below, and I hope that the author would find them useful.

1. I think that some more elaboration on why exposure to Soviet communism would be relevant to vaccine hesitancy is warranted. Is it something about the history of those countries? Is it something about the ideology? Culture? For example, recent research by Maaravi and colleagues shows the relationship between individualism / collectivism and COVID-19 spread (which I can only assume is correlated with exposure to Soviet communism, although they show a trend that to some extent is opposite to the current research results). Other unpublished research by Adam-Troian and colleagues (see here https://psyarxiv.com/nzg7x) examines the association between collectivism, masculinity (controlling for other relevant variables such as Human Development Index) to belief in conspiracy theories (which was linked to institutional trust and vaccine hesitancy). All of this to say that I think that some more precision about what is it exactly in exposure to Soviet communism that leads to vaccine hesitancy and lower levels of institutional and governmental trust is important.

R: Thank you for pointing out different possible mechanisms explaining our findings and the references. We have revised the two suggestions and expanded the introduction on pages 3-4 as well as the discussion on page 10. We link the exposure to Soviet communism effects to the experiences of enforced vaccinations common to all countries in the Soviet bloc, however we are unable to test nor rule out conclusively other potential mechanisms such as collectivist, individualistic or masculinity values. 

2. I believe that in order for the analysis to be persuasive it has to control for other relevant variables that are related to institutional, governmental and vaccine trust (both on the country and individual levels). Both the work by Adam-Troian et al. and Maaravi et al. I mentioned above control for some relevant such measures.

R: Thank you very much for the very relevant references and methodological suggestion. We have cited appropriately both studies in pages 3 and 10 and linked their results to our conclusions. Regarding the use of the control variables they use, please note our specification includes both country fixed-effects, which accounts for both country level time-invariant observable and unobservable differences. The inclusion of country level control variables as in the studies mentioned would create in our setting a problem of perfect multicolinearity. We included also additional individual controls to find robust results reported in Figure A1 of the Appendix. We believe however that since the measure of exposure to communism is exogenous (determined by year and place of birth) the inclusion of individual level attitudes or beliefs could lead to a problem of "bad control" as defined in Angrist and Pischke (2008). Hence, the former robustness check of various country sub-samples is better suited to our analysis and as such we place it in the main body of the paper.

Angrist, Joshua D., and Jörn-Steffen Pischke. Mostly harmless econometrics. Princeton University Press, 2008.

3. The results on “psychological mechanisms” go some way to address my first comment, but I think that there’s still a need to further explain what is it in the exposure to Soviet communism that drives these effects. Is it something about the type of regime? Is it something about the process in which these regimes collapsed and what happened since that broke individuals’ trust? Is the trend only relevant to Soviet communism? Why would that be the case? Is it because it collapsed? Is it just because vaccination was obligatory in the former Soviet Union? In other words, I think that more is needed to be done in order to explain whether the trends the authors report in the paper are specific to Soviet communism, and if so, why? Or whether it is something that has to do with regime changes or authoritarian regimes, etc.

R: We have now provided a more extensive and hopefully convincing explanation both in the Introduction and discussion sections referring to the general lack of transparency in public institutions, the power of Soviet authorities over individuals as well as military-like enforcement of vaccinations. The conspiracy and misinformation related to health care systems in Soviet times are associated not only with corruption but also with Soviet collectivism that paid little attention to individuals as well as with a denial of any danger in severe cases of environmental pollution, including radioactive contamination, as for example within first days after the accident in Chernobyl nuclear power plant.

4. As a minor comment, I think that it would be good to label the y-axis in the figures. I am not sure it is clear.

R: Thank you for this comment, we included it in our Figures which makes them much easier to read.

5. As another minor comment, maybe some more explanation about how exposure to communism was computed in the model. The authors indicate that it was measured in two different ways, but then it was less clear to me how these two different ways were combined.

R: Amended by providing on page 6 the general rule of setting the date of communism start to the date of Soviet constitution and of its end to the date of first free parliamentary (or presidential in the case of Russia) elections, and more details (specific dates separately for each country) in the Appendix.

Reviewer #2: When I saw the Abstract for this paper, I was very excited. I undertake sociological research on 'trust in vaccinations', so to see a potential paper exploring the impact of communism on trust in vaccinations was exciting. The authors use existing data from the Wellcome Global Monitor (which is fine) and undertake some statistical analysis (which I am not equipped to review). The Introduction is too broad, and does not adequately present and critique global literature on 'trust in vaccinations' and needs to engage much more deeply with the sociology of trust. There is a huge literature on trust in governments, medical institutions, pharmaceutical industries etc - very little if any of this has been engaged with. The authors need to be able to set their study and findings within these literatures - so they can identify 'what we already know' and 'what's new'. On lines 24-26, there's a referencing problem which goes throughout the paper (the first references are 6 and 1? In the Discussion, the authors could/should engage with sociologists who have explored how trust differs between different societies - for example, Fukuyama explored trust in different societal structures. In the Discussion, the paragraph on lines 151-161 does not seem to be relevant at all - the authors have introduced COVID-19 context, but the Wellcome data predated COVID-19.

R: We have expanded the literature on government trust in more detail and refer to the discussion in sociology. In addition to Simmel’s and Fukuyama’s seminal work on trust, we refer to more recent research on trust and confidence in public institutions in post-communist and other countries. 

Thank you for your point on the pre-COVID data. We explained why the attitudes towards vaccination in general might be relevant for the COVID-19 vaccine hesitancy on page 3. We have dealt with the referencing problem as well. 

Reviewer #3: Title: The institutional origin of vaccine distrust: Evidence from former-communist countries

Journal: Plos One

The article is a study of vaccine distrust in former-communist countries. Based on an analysis of data from the Welcome Global Monitor (WGM) database, the authors seek to show that exposure to communist regimes play an important role in distrust in public institutions that leads to vaccine hesitancy. Considering the scale of the challenge associated with vaccine hesitancy, especially during a pandemic, this article’s contributions to policy framing and guidelines are timely and necessary.

While the article does offer an important and timely exploration of vaccine hesitancy in former-communist countries, there is still some work to be done in establishing a coherent narrative, and tightening the arguments by addressing not only the results of the analysis but also more of the literature on trust, hesitancy and health policy. Addressing these issues (and a few other minor comments) will strengthen the arguments made in the article. More specific comments follow:

1. The authors provide very clear and admirable inclusion/exclusion criteria for which they should be congratulated. However, the grouping of former-communist counties and the assumption that all communist experiences are similar might be problematic. While it might be difficult to account for the broad variety of experiences based on the available database, I suggest the authors address this methodological difficulty.

R: We have done a number robustness check looking at the different types of communism, namely Soviet Union members, Russia, and selected Soviet Republics with strongest links to Western Europe, as presented in Subsection 3.3 Robustness on page 9. 

2. On page 3, the authors claim that the “military-like organizations and its compulsory participation lead to a questioning of such campaigns after transition”. While these claims might be intuitive to scholars of former-communist health systems, many of the journal’s readers may not be acquainted with this argument. Therefore, it would be helpful to add sources baking the claim.

R: Thank you for noting that, we have added additional references to the literature that documents these Soviet-specific practices.

3. In the Data section (page 4) the authors address the length of the influence of life under communist regimes. If possible, it would be helpful to address inter-generational effects as well.

R: We have discussed the role of intergenerational transmission of preferences, which is likely to take place as far as trust is concerned. However, if the preferences were not transmitted over generations and individuals unexposed to communism (or exposed to communism for relatively short period of time if we examine extensive margin of the exposure) in the formerly communist countries did not inherit any attitudes towards vaccination nor mistrust from their parents exposed to communism (for relatively long periods of time), the estimated effects of the exposure to communism should be even stronger. Therefore, the intergenerational transmission is not a problem in the study as it would result to the in the underestimation of the effects of the exposure to communism.

The fact that we find an effect on the intensive margin and not only on the extensive margin of the exposure suggests that documented effects are not driven by the transition but indeed by the exposure to communism. Despite our results support the trust mechanism, but we cannot conclusively rule out other mechanisms or say whether other authoritarian regimes or forms of communism would have a similar effect. Despite Soviet communism as one of authoritarian regimes shares a number of similarities with other communist regimes (such as for example forced vaccination in China), remains distinct from other authoritarian regimes. Our results are consistent with trust playing an important role, but we cannot test in the present study whether other forms of authoritarian regimes or forms of communism could have the same effect. 

4. In their analysis of Underlying mechanisms (page 6) the authors claim that trust has deteriorated. This framing assumes that trust existed in the past, however it’s unclear in the text if this is indeed the case.

R: We have clarified this point and elaborated more on the underlying mechanisms in Subsection 3.2. 

5. The article has the potential to contribute greatly to the discussion on trust in state institutions and vaccine hesitancy. However, in order to do so more attention is needed to the already broad body of knowledge on the issue. The theorizing of trust and hesitancy appears only in the discussion, and it would be helpful to address it earlier in the article.

R: We have added more on trust and vaccine hesitancy, and especially more theory in the earlier parts of the paper. 

6. Along the same line, a clearly framed discussion on trust in institutions and hesitancy would help the authors take their place in a very active field (and explain how is this case different from other cases).

R: We have improved the discussion.

7. I suggest the authors look at Heidi Larson’s book Stuck (2020) on vaccine rumours. While other works by Larson are cited, this might be an interesting read in terms of framing the context.

R: Thank you, we have included it.

8. On page 8, the authors shift to COVID-19. While this is obviously a burning issue (and of relevance to the article) it is not necessarily clear how a study conducted pre-COVID-19 relates to the matter at hand. A clearer connection should be made in order for this important section to be a better fit.

R: We have added references showing that hesitancy to other vaccines correlated with COVID-19 too. 

9. The authors end with a very interesting note on the use of compulsory mechanisms. I encourage the authors to be even bolder in their conclusions, and odd their recommendations.

R: We have sharpened our conclusions, thank you for encouraging us to do so.

10. As someone who is interested in trust in the state and health policy I have truly enjoyed reading this article. I thank the authors for their analysis.

R: Many thanks for reading our paper, we feel very honoured with your remark.

---

## [Decision Letter · Decision Letter 1]

5 Jan 2023

PONE-D-21-12657R1The institutional origins of vaccines distrust: Evidence from former-communist countriesPLOS ONE

Dear Dr. Nicińska,

Thank you for submitting your manuscript to PLOS ONE. After careful consideration, we feel that it has merit but does not fully meet PLOS ONE’s publication criteria as it currently stands. Therefore, we invite you to submit a revised version of the manuscript that addresses the points raised during the review process.

We look forward to receiving your revised manuscript.

Kind regards,

Jerg Gutmann

Academic Editor

PLOS ONE

Additional Editor Comments:

I have very recently taken over this submission as the academic editor. Previous reviewers have been approached, but not all of them have been able or willing to review the revised version of the article. The article has, thus, also been sent to one new reviewer who is an expert in the field and he or she very swiftly produced a review of the revised and resubmitted version of the article.

Two previous reviewers are satisfied with how their comments have been implemented. The new reviewer suggests a number of revisions before the article can be considered for publication. I agree with the reviewer's judgment that the authors currently seem to be leaving out essential information for being able to evaluate their empirical analysis. Also, some decisions need to be (better) justified. I would, thus, urge the authors to take the new reviewer's comments very seriously in revising the article. Should you be willing to revise the article based on these comments, the revised version would only be sent back to the new reviewer.

Reviewers' comments:

Reviewer's Responses to Questions

**Comments to the Author**

1. If the authors have adequately addressed your comments raised in a previous round of review and you feel that this manuscript is now acceptable for publication, you may indicate that here to bypass the “Comments to the Author” section, enter your conflict of interest statement in the “Confidential to Editor” section, and submit your "Accept" recommendation.

Reviewer #1: All comments have been addressed

Reviewer #2: All comments have been addressed

Reviewer #4: (No Response)

2. Is the manuscript technically sound, and do the data support the conclusions?

Reviewer #1: Yes

Reviewer #2: Yes

Reviewer #4: Partly

3. Has the statistical analysis been performed appropriately and rigorously? 

Reviewer #1: Yes

Reviewer #2: I Don't Know

Reviewer #4: No

4. Have the authors made all data underlying the findings in their manuscript fully available?

Reviewer #1: Yes

Reviewer #2: Yes

Reviewer #4: Yes

5. Is the manuscript presented in an intelligible fashion and written in standard English?

Reviewer #1: Yes

Reviewer #2: Yes

Reviewer #4: Yes

6. Review Comments to the Author

Reviewer #1: (No Response)

Reviewer #2: The authors have adequately addressed my concerns from my original review. They have engaged with some of the sociology of trust and more empirical research around trust in vaccinations

Reviewer #4: Although I wasn’t involved in the previous review of the manuscript, I am familiar with the work from a preprint. I generally like the work, but I have a few points that I think are worth considering before acceptance. Some are minor, while others are more major.

In the abstract it states “… from a long list of world countries…”, but I think there is no need to be so vague. I’d recommend stating the exact number. Also in the abstract, there is causal language without any causal tests to justify it: “past exposure to Soviet communism reduces trust in vaccinations”. I think based on the correlational nature of the data, a more accurate statement would be something like “past exposure to Soviet communism is associated with reduced trust in vaccinations”. I would tone back any causal language that occurs throughout the manuscript to be more inline with the nature of the data.

In the introduction, it was mentioned that experiencing mass vaccinations in Soviet countries might lead to vaccine distrust. This is perhaps true, but voluntary vaccination rates for the flu are considerably higher in former communist East Germany compared to West Germany (Rehmet et al., 2002), which might suggest it is, at least, more complicated.

Based on the argument presented in the manuscript, I was partly expecting a mediational analysis of the effect of communism on vaccination hesitancy. Namely, that it is mediated by lack of (some form of) trust. If such an analysis isn’t done, then talks about mechanisms should be more speculative. Recently published work that found trust in government mediated the association between former communist federal states and COVID-19 vaccination rates within Germany (at the state level rather than individual level; Martens, 2022). This is a highly relevant article for the current work.

I believe the results are presented in a more confusing manner than they need be, and they generally lack detail. What analyses were specifically run is not clearly presented. I assume these are a series of regressions, but then what are the exact results (e.g., B, CI, etc.)? I see figures, but tables with specific values would provide more information and aid in reproducing the effects and in any metanalysis down the road, etc. Consequently, although I generally like what the manuscript attempts to do, I feel unable to fully judge it because I am not confident in precisely what analyses were run or what was found.

I think there should be more mention of religion or spirituality since we know it influences vaccination skepticism (e.g., Rutjens et al., 2021) as well as COVID-19 vaccination rates (Martens & Rutjens, 2022). Given that the Soviet Bloc is predominantly Orthodox Christian, how can we be sure this isn’t a religion effect rather than a Soviet communism effect? I don’t actually think it is since Martens (2022) found a communist effect in Asia, but I believe this should at least be addressed in text as it is a likely contributing factor to vaccination hesitancy. Perhaps as a future direction.

What are the hypotheses precisely? In the introduction (sentences 5-9) it states that the paper examines the reaction to historical events (mass vaccination during Soviet times), weak trust in government and health system, suspicion of large business organizations (big pharma), and a reaction to egalitarian values of Soviet communism. Presumably how all of those are related to vaccine hesitancy (or vaccine trust as it is sometimes put). At other times the manuscript states it inquires about the effects of exposure to Soviet communism on trust in vaccines efficiency and safety (21-22), and whether communism explains different dimensions of vaccine trust (23-24). The first results presented give a breakdown of vaccine safety and efficiency (via a table) with some mention of communism. It next presents data to suggest an underlying mechanism (trust in government, doctors, etc.). From what I can tell, the section on underlying mechanisms is not actually testing underlying mechanisms, but rather an association consistent with the mechanism. Perhaps I’ve misinterpreted these results, but I believe that is partly part of the problem, since what was done is not clearly communicated. In addition, the distinction between intensive and extensive margin effects is interesting, but similar to above, it is difficult to follow what was precisely done here. What data/test actually supports that claim? The last part of the results is on robustness of the association. I see no data or results measuring mass vaccinations during Soviet times, suspicion about large businesses, or egalitarian values, so contrary to the introduction, these were not evaluated. They might be aspects that support your hypotheses (which I take to be that Soviet communism in particular leads to lack of trust in a variety of areas, which then leads to vaccine hesitancy), but they aren’t part of the data collection or analyses.

Generally speaking, I found the introduction and discussion to not be overly developed. I think sub-headings with clear points (perhaps framed around hypotheses) could help here. That might also help the results section (i.e., frame results more explicitly around hypotheses testing).

It isn’t clear to me why the focus is solely on Soviet communism. Is it because of mass vaccination efforts in Soviet communism countries that didn’t happen in Asian/African communist countries? Asian communism is associated with lower vaccinations rates for COVID-19 (Martens, 2022), so perhaps it isn’t something unique to Soviet communism. If it is something unique to Soviet communism then you would expect an effect with Soviet communism but not other forms of communism, but you can test this by including other forms of communism in the analyses. If running this test isn’t possible for some reason, then I’d like to see more of an expansion on justifying why Soviet regions are the focus.

Trust in vaccinations seems to be operationalized as both vaccine efficiency and safety. I’m not convinced that those are two measures of vaccine trust. Maybe they do (which would be good to hear justification for), but someone might have complete trust in taking a vaccine but think it is not efficient (e.g., it takes multiple injections for full immunity). Safety appears to be more directly about trust than efficiency, but neither are the most direct, which would be something like “I trust vaccines”, which unfortunately doesn’t appear to be part of the dataset. I’d recommend rethinking the label (hesitancy was used earlier in the paper, so might be used for consistency, but views/opinion could also work here. Skepticism is another possibility, and previous work that used the same data set called those variables skepticism; Martens & Rutjens, 2022) and/or justifying how those are actual measures of trust. I assume those two measures are highly correlated, which would be good to see since they are the main DVs. It might make sense to combine them into a scale which might help with any odd effects of any individual item, unless the authors are trying to make an argument more specifically about each aspect (efficiency and safety). I take it that this is more about hesitancy, so a scale might make sense here.

The authors state, “In our multivariate analysis” (138), but what type of multivariate analysis precisely? A few more details here would be quite useful. It would also be useful to put some statistics in text or at least a table with all the relevant values. Similarly, the authors state that the effect sizes depend on exposure to communism, but the min/max values here would be useful to know in text and also help with interpreting the data. More details are needed all around.

Table 1 is used to suggest that Soviet countries have lower vaccine efficiency and safety, but the countries are sorted by alphabetical order rather communism, so this pattern is difficult to see in the table. I’d recommend breaking it down with Soviet countries on one side and non-Soviet on the other so that the comparisons are much more apparent. Some descriptive statistics would also help here (Ms, SDs, etc.). The authors seem to have gone for a minimalistic presentation of the data, but the lack of details raises more questions that might easily be answered by including more details in text, in tables, etc.

Martens, JP (2022). Communism’s lasting effect? Former communist states and COVID-19 vaccinations. Cross-Cultural Research, https://doi.org/10.1177/10693971221134181

Martens, JP, & Rutjens, BT (2022). Spirituality and religiosity contribute to ongoing COVID-19 vaccination rates: comparing 195 regions around the world. Vaccine: X, https://doi.org/10.1016/j.jvacx.2022.100241

Rutjens, B. T., Sengupta, N., der Lee, R. van, van Koningsbruggen, G. M., Martens, J. P., Rabelo, A., & Sutton, R. M. (2022). Science Skepticism Across 24 Countries. Social Psychological and Personality Science, 13(1), 102–117. https://doi.org/10.1177/19485506211001329

Rehmet, S., Ammon, A., Pfaff, G., Bocter, N., & Petersen, L. R. (2002). Cross-Sectional study on influenza vaccination, Germany, 1999–2000. Emerging Infectious Diseases, 8(12), 1442–1447. https://doi.org/10.3201/eid0812.010497

7. PLOS authors have the option to publish the peer review history of their article (what does this mean?). If published, this will include your full peer review and any attached files.

Reviewer #1: No

Reviewer #2: No

Reviewer #4: No

---

## [Author Response · Author response to Decision Letter 1]

24 Jan 2023

Response to the reviews

In the abstract it states "… from a long list of world countries…", but I think there is no need to be so vague. I'd recommend stating the exact number. Also in the abstract, there is causal language without any causal tests to justify it: "past exposure to Soviet communism reduces trust in vaccinations". I think based on the correlational nature of the data, a more accurate statement would be something like "past exposure to Soviet communism is associated with reduced trust in vaccinations". I would tone back any causal language that occurs throughout the manuscript to be more in line with the nature of the data.

Response: Thank you for this important comment, we have amended the language to be more precise in terms of the nature of our findings throughput the entire article. We also provided the exact number of examined countries (122) in the Abstract and in the Data description (page 5).

In the introduction, it was mentioned that experiencing mass vaccinations in Soviet countries might lead to vaccine distrust. This is perhaps true, but voluntary vaccination rates for the flu are considerably higher in former communist East Germany compared to West Germany (Rehmet et al., 2002), which might suggest it is, at least, more complicated.

Response: We discuss it the revised draft referring to the evidence suggesting difference between East and West Germany in Rehmet et al. (2002) (page 10). A stated in the paper, the difference arises due to a higher access to family physicians in the East Germany, which are mainly the ones making vaccinations recommendations, given that there is no need of GP referral in Germany to see a specialist. Moreover, we include the ECDC evidence for seasonal flu suggesting that, as we argue, eastern European countries exhibit the lowest rates of influenza vaccination among olde people and pregnant women (ECDC, 2023): https://www.ecdc.europa.eu/en/publications-data/seasonal-influenza-vaccination-antiviral-use-eu-eea-member-states.

Based on the argument presented in the manuscript, I was partly expecting a mediational analysis of the effect of communism on vaccination hesitancy. Namely, that it is mediated by lack of (some form

of) trust. If such an analysis isn't done, then talks about mechanisms should be more speculative. Recently published work that found trust in government mediated the association between former communist federal states and COVID-19 vaccination rates within Germany (at the state level rather than individual level; Martens, 2022). This is a highly relevant article for the current work.

Response: Thanks for the reference. We now cite Martens (2022) in the updated version of the manuscript when discussing potential hypotheses that might be explaining the association in the Introduction (page 5).

Regarding the evidence on mechanisms, I believe this is probably a confusion by the different use of the word across disciplines. Typically, the analysis of mechanisms in economic papers is based on heterogeneous effects or evidence supporting a particular hypothesis, rather than conducting mediational analysis, which is a technique rarely used in our discipline. As the reviewer states, in our paper, our section on mechanism shows that Soviet communism reduced trust in these institutions, which is consistent with the hypothesis that trust could be driving the negative association between exposure to communism and trust in vaccine effectiveness and safety. 

We understand however the section might create confusion to colleagues of other disciplines and therefore we have toned down the results of the section and avoid the use of the term analysis of mechanism when presenting the results on trust. We hope the section is better communicated now.

I believe the results are presented in a more confusing manner than they need be, and they generally lack detail. What analyses were specifically run is not clearly presented. I assume these are a series of regressions, but then what are the exact results (e.g., B, CI, etc.)? I see figures, but tables with specific values would provide more information and aid in reproducing the effects and in any metanalysis down the road, etc. Consequently, although I generally like what the manuscript attempts to do, I feel unable to fully judge it because I am not confident in precisely what analyses were run or what was found.

Response: Thank you for this comment, we have revised the description of the methods in the Identification Strategy section (page 7) to make it clearer. We run two separate OLS estimations for the trust in vaccines’ efficiency and safety, using either dichotomous or continuous measure of exposure to communism as the explanatory variable, controlling for fixed effects of birth year, country of birth as well as for the country-specific time trends. In addition, we control for gender and a number of other controls in the robustness tests. 

We have chosen to present the results in the graphic form so that they are more communicative. The point estimates are shown along with the Confidence Intervals at 95% significance level, and the statistical significance of the estimate is shown using asterisks by the point estimate. The same amount of information presented in a table would be less readable, thus we decided to stay with the Figures in the main body of the paper and provide all the details in the following Tables added to the appendix. We have also revised the Notes to all Figures to make sure all these information are a part of each Figure.

Table S3 Effects of the Exposure to Soviet Communism on Trust in Vaccines’ Efficiency and Safety.

 Vaccine efficiency Vaccine efficiency

Exposure (continuous) -0.03820*** -0.03984***

 (0.0139) (0.0152)

N 107868 107745

R2 0.100 0.143

Exposure (dummy) 

-0.07132** 

-0.08504**

 (0.0273) (0.0357)

N 107868 107745

R2 0.100 0.143

Age-fixed effect Yes Yes

Country-fixed effect Yes Yes

Country-specific time trend Yes Yes

Notes: The continuous measure of exposure is the length of exposure to Soviet communism in years smoothed with inverse hyperbolic sine function. The dummy measure of exposure is the having any exposure to Soviet communism. The continuous measure of exposure is the length of exposure to Soviet communism in years smoothed with inverse hyperbolic sine function. Standard errors clustered by country. Statistical significance: * – p < 0.10, ** – p < 0.05, *** – p < 0.01.

Source: WGM 2018

Table S4 Effects of the Exposure to Soviet Communism on Generalized Trust and Trust in Medical Care.

 Government Health Advice Medical Advice 

from Doctors Medical Personnel Hospital and 

Health Clinics Government People in Neighbourhood

Exposure (continuous) -0.04838*** -0.03662*** -0.04108*** -0.01307*** -0.05669*** -0.02178**

 (0.0128) (0.0095) (0.0097) (0.0049) (0.0154) (0.0106)

N 115213 119351 121297 118357 112221 120725

R2 0.146 0.126 0.139 0.099 0.209 0.140

Exposure (dummy) -0.07932*** -0.05639*** -0.06273*** -0.01480 -0.09586*** -0.04130**

 (0.0282) (0.0185) (0.0206) (0.0117) (0.0319) (0.0208)

N 115213 119351 121297 118357 112221 120725

R2 0.145 0.126 0.139 0.099 0.209 0.140

Age-fixed effect Yes Yes Yes Yes Yes Yes

Country-fixed effect Yes Yes Yes Yes Yes Yes

Country-specific time trend Yes Yes Yes Yes Yes Yes

Notes: The continuous measure of exposure is the length of exposure to Soviet communism in years smoothed with inverse hyperbolic sine function. The dummy measure of exposure is the having any exposure to Soviet communism. The continuous measure of exposure is the length of exposure to Soviet communism in years smoothed with inverse hyperbolic sine function. Standard errors clustered by country. Statistical significance: * – p < 0.10, ** – p < 0.05, *** – p < 0.01.

Source: WGM 2018

I think there should be more mention of religion or spirituality since we know it influences vaccination skepticism (e.g., Rutjens et al., 2021) as well as COVID-19 vaccination rates (Martens & Rutjens, 2022). Given that the Soviet Bloc is predominantly Orthodox Christian, how can we be sure this isn't a religion effect rather than a Soviet communism effect? I don't actually think it is since Martens (2022) found a communist effect in Asia, but I believe this should at least be addressed in text as it is a likely contributing factor to vaccination hesitancy. Perhaps as a future direction.

Response: Thanks for the references. We have added to the introduction a reference to religion or spirituality as factors that may also affect vaccinations rate and cite the mentioned studies. Regarding whether the effect could be driven by Orthodox Christian, the use of country specific fixed effects lowers this concern. Moreover, we control for individual religiosity in the robustness checks to find reassuring results. Nonetheless, we test this hypothesis in the updated version of the manuscript (Figure S2) through exploring whether the association is stronger across Orthodox and non-Orthodox Soviet countries. We do not find any significant differences, as the coefficients on the interaction of the exposures (both continuous and dichotomous) with Orthodox country shown in the Table below are statistically indistinguishable from zero. 

Table R1 Robustness to the Orthodox Christian Country.

 Vaccine efficiency Vaccine efficiency

Exposure (continuous) × Orthodox -0.04404

(0.0280) 0.04090

(0.0304)

N

R2 11186

0.065 11151

0.078

Exposure (dummy) × Orthodox -0.08267

(0.0619) 0.04828

(0.0772)

N 

R2 11186

0.065 11151

0.078

Age-fixed effect Yes Yes

Country-fixed effect Yes Yes

Country-specific time trend Yes Yes

Notes: The countries classified as predominantly Orthodox Christian are: Belarus, Bulgaria, Cyprus, Georgia, Greece, Macedonia, Moldova, Montenegro, Romania, Russia, Serbia, Ukraine. The continuous measure of Exposure to communism is the length of exposure to Soviet communism in years smoothed with inverse hyperbolic sine function. Standard errors clustered by country. Statistical significance: * – p < 0.10, ** – p < 0.05, *** – p < 0.01.

Source: WGM 2018.

What are the hypotheses precisely? In the introduction (sentences 5-9) it states that the paper examines the reaction to historical events (mass vaccination during Soviet times), weak trust in government and health system, suspicion of large business organizations (big pharma), and a reaction to egalitarian values of Soviet communism. Presumably how all of those are related to vaccine hesitancy (or vaccine trust as it is sometimes put). At other times the manuscript states it inquires about the effects of exposure to Soviet communism on trust in vaccines efficiency and safety (21-22), and whether communism explains different dimensions of vaccine trust (23-24).

 The first results presented give a breakdown of vaccine safety and efficiency (via a table) with some mention of communism. 

Response: Thanks for this comment. We have clarified it in the article by rephrasing the sentences 5-9 in the introduction and providing explicit hypothesis on page 4. Our hypothesis is that exposure to communism is associated vaccine scepticism i.e. reduced trust both in vaccines’ safety and efficiency. The rationale for the hypothesis are the characteristics of the exposure to communism and experiences it entailed, specifically (forced) mass vaccination, generalized trust deterioration, reduced confidence in public institutions, health care system rooted in the Semashko model, prevalence of egalitarian values as well as suspicion of large business organization. These are the traits documented in the literature discussed and cited in the Introduction.

It next presents data to suggest an underlying mechanism (trust in government, doctors, etc.). From what I can tell, the section on underlying mechanisms is not actually testing underlying mechanisms, but rather an association consistent with the mechanism. Perhaps I've misinterpreted these results, but I believe that is partly part of the problem, since what was done is not clearly communicated

Response: We believe this is probably a confusion by the different use of the word across disciplines. Typically, the analysis of mechanisms in economic papers is based on heterogeneous effects or evidence supporting a particular hypothesis, rather than conducting mediational analysis, which is a technique rarely used in our discipline. As the reviewer states, in our paper, our section on mechanism shows that Soviet communism reduced trust in these institutions, which is consistent with the hypothesis that trust could be driving the negative association between exposure to communism and trust in vaccine effectiveness and safety. 

We understand however the section might create confusion to colleagues of other disciplines and therefore we have toned down the results of the section and avoid the use of the term analysis of mechanism when presenting the results on trust. We hope the section is better communicated now.

In addition, the distinction between intensive and extensive margin effects is interesting, but similar to above, it is difficult to follow what was precisely done here. 

Response: Thanks for the comment. Intensive and extensive margins are common terms in economics. In this case, the extensive margin of exposure to communist is a variable that takes the value of 1 if the individual is exposed to communism and 0 otherwise. The intensive margin measures the number of years of exposure of the individual to Soviet communism.

We understand PLOSONE is an interdisciplinary journal and therefore, we have replaced these terms in the updated version of the paper. Rather than using extensive and intensive margin of exposure to communism we have used the terms “exposed to communism (0-1)” and “number of years one was exposed to communism (0-70 years)”. 

The last part of the results is on robustness of the association. I see no data or results measuring mass vaccinations during Soviet times, suspicion about large businesses, or egalitarian values, so contrary to the introduction, these were not evaluated. They might be aspects that support your hypotheses (which I take to be that Soviet communism in particular leads to lack of trust in a variety of areas, which then leads to vaccine hesitancy), but they aren't part of the data collection or analyses.

Response: Thanks for this point, we have now stated the hypothesis and clarified what can be tested empirically, based on the available data. We are able to test a number of channels through which communism might have affected (e.g. Trust in medical advice from doctors, in medical personnel, government, etc.). We recognize that there are other potential attitudes we do not measure such as ‘suspicion about large businesses’, or ‘egalitarian values’ documented in the literature. In the revised article we make clear distinction between findings based on our own empirical analysis and where we rely on existing published evidence. 

Generally speaking, I found the introduction and discussion to not be overly developed. I think sub-headings with clear points (perhaps framed around hypotheses) could help here. That might also help the results section (i.e., frame results more explicitly around hypotheses testing).

Response: Thanks for this suggestion! We have expanded the discussion and organised the results and discussion with subheadings now. 

It isn't clear to me why the focus is solely on Soviet communism. Is it because of mass vaccination efforts in Soviet communism countries that didn't happen in Asian/African communist countries? Asian communism is associated with lower vaccinations rates for COVID-19 (Martens, 2022), so perhaps it isn't something unique to Soviet communism. If it is something unique to Soviet communism then you would expect an effect with Soviet communism but not other forms of communism, but you can test this by including other forms of communism in the analyses. If running this test isn't possible for some reason, then I'd like to see more of an expansion on justifying why Soviet regions are the focus.

Response: Thanks for the comment, we clarify that in the paper on page 5. We focused on Soviet communism for both practical and theoretical reasons. Firstly, for Soviet countries we could identify a relatively common treatment and transition period, as well as a homogeneous sample of what is a communist country. The inclusion of countries such as China or Vietnam, formally communist countries but with economic systems that have transitioned into capitalism on many aspects, would complicate the interpretation of the treatment effects. Similarly, the inclusion of some African and Asian countries that experienced for shorter periods different forms of communism (typically associated with civil wars) and transitioned into right-wing dictatorships could difficult the interpretation of the results. Thus, through restricting the analysis to Soviet countries, which experienced a relatively common political and economic system with clear dates for the beginning and end of communism, and then, similar economic and political systems during transition, we could yield more homogeneous and easier to interpret results. Secondly, gathering precise and comprehensive information on dates of the beginning and end of communism in every world country that experienced communism would require resources that we do not have at the moment. Finally, it is worth mentioning we have mentioned this point in the text now. 

Trust in vaccinations seems to be operationalized as both vaccine efficiency and safety. I'm not convinced that those are two measures of vaccine trust. Maybe they do (which would be good to hear justification for), but someone might have complete trust in taking a vaccine but think it is not efficient (e.g., it takes multiple injections for full immunity). Safety appears to be more directly about trust than efficiency, but neither are the most direct, which would be something like "I trust vaccines", which unfortunately doesn't appear to be part of the dataset. I'd recommend rethinking the label (hesitancy was used earlier in the paper, so might be used for consistency, but views/opinion could also work here. Skepticism is another possibility, and previous work that used the same data set called those variables skepticism; Martens & Rutjens, 2022) and/or justifying how those are actual measures of trust.

Response: Thanks for this helpful suggestion. We have revised the article accordingly labelling examined beliefs as vaccine scepticism, and clearly state that we operationalize it by the two measures: of trust in vaccines efficiency and safety. We have expanded the points regarding vaccine hesitancy and scepticism of this in the paper, and more specifically mentioned that they refer to instrumental dimensions of trust which have been reported as the key variable in a number of studies. 

I assume those two measures are highly correlated, which would be good to see since they are the main DVs. It might make sense to combine them into a scale which might help with any odd effects of any individual item, unless the authors are trying to make an argument more specifically about each aspect (efficiency and safety). I take it that this is more about hesitancy, so a scale might make sense here.

Response: Thank you for this point. We have revised the text making sure that the two outcomes are interpreted separately, and the distinction between them is pronounced enough. 

We have discussed the selection of the dependent variable at the very start of the research, and considered generating an index or a scale. We concluded these considerations with the decision to use the variables directly reported directly by the respondents for two main reasons.

First, we are not experts in the development of scales or indexes, but we are aware of the methodological complexities related to the development of an accurate and robust measures combining a number of variables, and one of them is free from limitations. We decided that each method has limitations reducing the conclusiveness of our empirical analysis in comparison to the simple analysis that we used as our main result. 

Second, the interpretation of results using the exact questions respondents were asked in the survey is straightforward and accurate without any additional assumptions. As economists, we are hesitant to use any variable that is not directly observed in the data as the dependent variable, because of the challenge with proper interpretation of the results. 

The authors state, "In our multivariate analysis" (138), but what type of multivariate analysis precisely? A few more details here would be quite useful. It would also be useful to put some statistics in text or at least a table with all the relevant values. Similarly, the authors state that the effect sizes depend on exposure to communism, but the min/max values here would be useful to know in text and also help with interpreting the data. More details are needed all around.

Response: Thank you for this comment, we have revised the description of the methods in the Identification Strategy section (page 7) to make it clearer. We have also improved the presentation of the mathematical equation representing the estimation models: we run two separate OLS estimations for the trust in vaccines’ efficiency and safety, using either dichotomous or continuous measure of exposure to communism as the explanatory variable, controlling for fixed effects of birth year, country of birth as well as for the country-specific time trends. In addition, we control for gender and a number of other controls in the robustness tests. Specifically, we add to the set of controls sequentially: dummy for living in the urban of rural area, dummy for having children, dummy for individual religiosity, and education level (primary, secondary, tertiary). On top of that, we have added the interaction term of the treatment variable (exposure to Soviet communism) with Orthodox country, as suggested above. 

The table S1 with descriptive statistics of variables used in the empirical analysis was slightly reorganized and provides details concerned with: the mean, minimum and maximum values, as well as the standard deviation and the number of non-missing observations.

In the revised paper we pay more attention to the quantitative interpretation of result.

Table 1 is used to suggest that Soviet countries have lower vaccine efficiency and safety, but the countries are sorted by alphabetical order rather communism, so this pattern is difficult to see in the table. I'd recommend breaking it down with Soviet countries on one side and non-Soviet on the other so that the comparisons are much more apparent. Some descriptive statistics would also help here (Ms, SDs, etc.). The authors seem to have gone for a minimalistic presentation of the data, but the lack of details raises more questions that might easily be answered by including more details in text, in tables, etc.

Response: Thanks for this point, we have decided to present these data graphically, using maps (Fig 1) available in public domain. Thanks to that, the difference between countries from the Soviet bloc and the rest of the world can be spotted much easier.

---

## [Editor Report · Decision Letter 2]

8 Feb 2023

PONE-D-21-12657R2The institutional origins of vaccines distrust: Evidence from former-Soviet countriesPLOS ONE

Dear Dr. Nicińska,

Thank you for submitting your revised manuscript. Unfortunately, the reviewer who suggested a major revision of your manuscript was not available for private reasons to review your manuscript. Thus, I have checked the implementation of the suggestions myself and they appear convincing. Before we accept your manuscript for publication, I would only ask you to fix some minor mistakes in the manuscript:

- Your estimation equation suggests that year of birth fixed effects vary over time. That is not correct. These are individual specific attributes that do not vary over time. Please adjust the index in the equation - unless I am misinterpreting what you are doing.

- Figure 1: I assume that the color-categories are based on quartiles or quintiles? Please provide the information what logic the color coding is based on. Also, it does not seem to make sense that the white category refers to values from 0 to 0!? Are these missing values? Please check and correct if necessary. (I assume this must be a mistake).- Figure 5: Please use only standard-English terms to label your figures. You can also drop the labels on the vertical axes to be consistent.

- All figures and tables (including supplementary material): I would strongly suggest to consistently use the same number of digits after the decimal point - unless there are reasons to deviate. Anything else looks very unprofessional. Also, showing about 8 digits after the decimal point does not convey any meaningful or comprehensible information, so I would suggest to limit the number of digits to a reasonable amount while still being sufficiently precise.- You may also give the manuscript another read for grammar and other language mistakes. If you make any other changes to the manuscript, these should, however, be communicated.

We look forward to receiving your revised manuscript.

Kind regards,

Jerg Gutmann

Academic Editor

PLOS ONE
---

## [Author Response · Author response to Decision Letter 2]

13 Feb 2023

Dear Editor,

Thank you for quick response to our resubmission. 

Please find our response to your comments

- Your estimation equation suggests that year of birth fixed effects vary over time. That is not correct. These are individual specific attributes that do not vary over time. Please adjust the index in the equation - unless I am misinterpreting what you are doing.

Thank you very much for spotting this mistake, we have corrected the equation in the revised manuscript.

- Figure 1: I assume that the color-categories are based on quartiles or quintiles? Please provide the information what logic the color coding is based on. Also, it does not seem to make sense that the white category refers to values from 0 to 0!? Are these missing values? Please check and correct if necessary. (I assume this must be a mistake).

Thanks, we have refined Figure 1. Specifically, we explained the categories depicted, cleaned up the brackets in the legend, and explained that white polygons on the map represent areas with no data available. We show the values of trust in efficiency and safety of vaccines by quantile, as described in the legend to each map.

- Figure 5: Please use only standard-English terms to label your figures. You can also drop the labels on the vertical axes to be consistent.

Thanks, we have revised labels for the examined sub-groups of communist countries accordingly.

- All figures and tables (including supplementary material): I would strongly suggest to consistently use the same number of digits after the decimal point - unless there are reasons to deviate. Anything else looks very unprofessional. Also, showing about 8 digits after the decimal point does not convey any meaningful or comprehensible information, so I would suggest to limit the number of digits to a reasonable amount while still being sufficiently precise.

We appreciate this comment and agree, it was important to improve our Tables (and Figures), We use 2 decimal points in the main body of the paper, and decided to be more detailed (3 decimal points) in the Supplementary Information. 

- You may also give the manuscript another read for grammar and other language mistakes. If you make any other changes to the manuscript, these should, however, be communicated.

We have revised the language and grammar very carefully, correcting text in a number of places without introducing any major changes to the contents of the paper.

---

## [Editor Report · Decision Letter 3]

15 Feb 2023

The institutional origins of vaccines distrust: Evidence from former-Soviet countries

PONE-D-21-12657R3

Dear Dr. Nicińska,

We’re pleased to inform you that your manuscript has been judged scientifically suitable for publication and will be formally accepted for publication once it meets all outstanding technical requirements.

Kind regards,

Jerg Gutmann

Academic Editor

PLOS ONE
---

## [Editor Report · Acceptance letter]

20 Feb 2023

PONE-D-21-12657R3 

The institutional origins of vaccines distrust: Evidence from former-Soviet countries 

Dear Dr. Nicińska:

I'm pleased to inform you that your manuscript has been deemed suitable for publication in PLOS ONE. Congratulations! Your manuscript is now with our production department. 

Kind regards, 

on behalf of

Prof. Dr. Jerg Gutmann 

Academic Editor

PLOS ONE